# A Novel Dantrolene Sodium-Loaded Mixed Micelle Containing a Small Amount of Cremophor EL: Characterization, Stability, Safety and Pharmacokinetics

**DOI:** 10.3390/molecules24040728

**Published:** 2019-02-18

**Authors:** Wenzhen Jin, Xiaochuan Tan, Jin Wen, Ya Meng, Yujia Zhang, He Li, Dong Jiang, Hui Song, Wensheng Zheng

**Affiliations:** 1State Key Laboratory of Bioactive Substance and Function of Natural Medicines & Beijing Key laboratory of Drug Delivery Technology and Novel Formulation, Institute of Materia Medica, Chinese Academy of Medical Sciences & Peking Union Medical College, Beijing 100050, China; jinwenzhen@imm.ac.cn (W.J.); tan@imm.ac.cn (X.T.); 18801316516@163.com (Y.M.); zhyj@imm.ac.cn (Y.Z.); lihe@imm.ac.cn (H.L.); jiangdong@imm.ac.cn (D.J.); angussong@imm.ac.cn (H.S.); 2Chinese Pharmaceutical Association, Beijing 100022, China; jin_ann1463@163.com

**Keywords:** mixed micelle, dantrolene sodium, Cremophor EL, drug safety, pharmacokinetics

## Abstract

Dantrolene sodium (DS) is the only drug specifically used for the treatment of malignant hyperthermia. Nevertheless, its clinical application is significantly restricted due to its aqueous insolubility and the limited formulations available in clinical practice. In order to solve these problems, a novel mixed micelle composed of phospholipid and Cremophor EL was designed and evaluated. The mixed micelle was prepared using a stirring- ultrasonic method. The Dynamic Light Scattering (DLS) results showed that the micelle was small in size (12.14 nm) and narrowly distributed (PdI = 0.073). Transmission Electron Microscopy (TEM) images showed that the micelle was homogeneous and spherical. The stability study indicated that the system was stable for storage and dilution with distilled water, while the safety testing showed that the micelle was safe for intravenous administration with low hemolysis rates and low allergic reaction rates. In the pharmaceutics study, the C_max_ and AUC_0-t_ of the DS-loaded micelle were 4- and 4.5- folds higher than that of the DS. Therefore, the constructed phospholipid-Cremophor EL mixed micelle is a promising drug delivery system for DS.

## 1. Introduction

Malignant hyperthermia (MH) is a rare autosomal dominant pharmacogenetic syndrome, characterized by the mutation of sarcoplasmic reticulum Ca^2+^ release channel in skeletal muscle cells [1]. MH can be triggered by depolarizing muscle relaxants [2] and halogenated volatile anesthetics, such as halothane, enflurane and isoflurane, which may lead to a fatal hypermetabolic state. The clinical symptoms vary from mild to potentially lethal, including tachycardia, hypercapnia, hypoxaemia, muscle rigidity and metabolic acidosis. The incidence of malignant hyperthermia is between 1:10,000 and 1:250,000 [3,4]; nevertheless, it can affect all ethnic groups worldwide.

Dantrolene sodium (hydrated 1-(((5-(4-nitrophenyl)-2-furanyl)methylene)amino)-2,4-imidazol- idinedione sodium salt, DS, structural formula shown in Figure 1), which was initially developed as a muscle relaxant for long-term treatment of muscle spasticity, is the only specific drug available for the treatment of malignant hyperthermia. DS is a direct-acting skeletal muscle relaxant that exerts its effects by dissociating excitation-contraction coupling in muscle via inhibition of the release of calcium from intracellular storage sites within the sarcoplasmic reticulum. DS is metabolized in the liver by oxidative and reductive pathways; and most of the drug is metabolized through the oxidative pathway into 5-hydroxydantrolene [5]. DS can be administrated orally or intravenously. From the 1960s to this day, DS has been responsible for decreasing the mortality rates of MH from 80% to 10%, respectively [6]. Nevertheless, the pharmacological effects of DS are severely limited due to the poor solubility of 8.34 μM at pH 7.4, 25 °C [7]. Therefore, there is an urgent need to develop a proper delivery system that can improve the solubility and bioavailability of DS.

To date, numerous systems have been developed for insoluble drugs. Mixed micelles, as promising drug carriers, have attracted a lot of attention in recent years [8]. Mixed micelles consist of a hydrophobic block that enables the poorly soluble drugs to be incorporated as an internal core and a hydrophilic block as a surrounding shell. Consequently, the structure can enhance the solubilization of a hydrophobic drug in biological media, improving the bioavailability and stability of drugs [9]. The nanoscopic size (<100 nm) of micelles allows them to be easily transferred to the systemic circulation. In addition, mixed micelle has shown to possess a higher stability and drug-loading efficiency compared to micelles with the single components [10].

Suitable phospholipid–surfactant mixtures have recently gained lot of attention for their industrial-applied potential [11]. Different ratios of phospholipid and surfactant lead to different carriers, thus, micelles can only be prepared using the proper ratio. The stability and permeability of mixed micelles is influenced by the nature, as well as by the absolute and relative concentration of the surfactant used [12]. Phospholipids are nontoxic and biocompatible, which makes them suitable for intravenous preparations. Cremophor EL is a commonly used non-ionic surfactant with great emulsifying properties both in experiments and the pharmaceutical industry [13]. Therefore, a formulation of mixed micelle established with phospholipid and remophor EL may have a great potential for solubilization and stability enhancement. In this study, phospholipid-Cremophor EL mixed micellar system were developed and assayed as potential carriers for DS. The particle size, morphology, stability toward dilution and storage, and critical micelle concentration (CMC) were measured. The pharmacokinetics and safety of DS-loaded mixed micelles were also evaluated.

## 2. Results and Discussion

### 2.1. Preparation of DS-Loaded Micelle

A standard orthogonal array matrix L_9_ (3^4^) was constructed using three factors (the amount of phospholipid, Cremophor EL and ethanol) and three levels to select the optimum formulation. The polydispersity index (PdI) and the DS encapsulated in micelle were labeled as P and Ce, respectively, and W (W = Ce/p) was taken as the index. According to the orthogonal design, the optimal formulation DS-loaded micelle was obtained as follows: 20 mg of DS, 0.3 mL of phospholipid ethanol solution (250 mg/mL), 1.2 mL of Cremophor EL and 0.5 mL of anhydrous ethanol. The DS-loaded micelle was prepared for further analysis.

The selection of the delivery system is crucial for improving the solubility of drugs. Mixed micelles have been shown to possess a higher stability and drug-loading efficiency compared to polymeric micelles. However, the micelle preparation process, which includes the procedure of synthesis and structural modification of polymer is a complicated and time-consuming process [14]. The mixed micelles prepared with surfactants have all the advantages of polymeric micelles while avoiding the complex preparation process, making them more popular for practical applications. Phospholipid and Cremophor EL are the most common and universal surfactants applied in practice by the pharmaceutical industry. In our study, the system composed of phospholipid and Cremophor EL was cost-effective, convenient and easy to prepare. The introduction of phospholipid with great biocompatibility enhanced the stability, while the combined use of the phospholipid and Cremophor EL decreased the amount required for establishing an ideal micelle system.

### 2.2. Characterization of DS-M

The size and size distribution of micelle were measured by DLS, while their morphological features were observed by TEM.

The average size of the mixed micelles was 12.21 nm, with a PdI of 0.073 and a zeta potential of −18.4 mV (Figure 2A,B). Compared with drug-free micelle, the DS-loaded micelles had a larger size. In addition, the TEM images of different micelles were different. The blank micelle image (Figure 2C) and drug-loaded micelle image (Figure 2D) showed that the morphology of mixed micelle was homogeneous spherical and the size of micelle was small and narrowly distributed, which was in agreement with the results of particle size analysis. Moreover, the phospholipid micelle (Figure 2E) was of larger size. Besides, the EE of DS in the mixed micelle was 92.6%.

CMC is an important parameter that needs to be considered when investigating the stability of drug-loaded micelles, both in vitro and in vivo [15]. The CMC value of the micelles was measured using a pyrene-based fluorescent probe. The intensity ratio I_1_/I_3_ of the first (373 nm) and the third (384 nm) vibronic peaks of the pyrene fluorescence spectrum changed according to the concentration of micelle.

As shown in Figure 3A, the value of I_373_/I_384_ sharply decreased at a certain concentration because of the micelle formation. According to the curve, the CMC of the mixed micelles was 2.835 × 10^−4^% (*w*/*v* while the CMC of Cremophor EL was 0.009% (*w*/*v*). It was obvious that the CMC value of the micelles sharply decreased compared with the Cremophor EL data. Normally, a high and low I1/I3 ratio indicate a polar and nonpolar environment, respectively. The polarity ratio reached the lowest when the volume of Cremophor EL was 1.2 mL, indicating more micelle aggregation (Figure 3B). These data confirmed the fact that the combined use of phospholipid and Creomphor EL results in a stronger tendency to form micelles and increases the micelle stability.

### 2.3. Stability Study

In order to study the storage and dilution stability of the mixed micelles, we measured the size and EE changes. Regarding the storage stability, the size and EE of samples before and 7 days after storage at room temperature were measured and compared. As for the stability toward dilution, the size and appearance of samples diluted with distilled water, saline and 5% glucose solution, respectively, were also observed and compared. The data are shown in Figure 3C. Briefly, almost no changes in size and a slight decrease in EE were observed before and after storage, regardless of temperature; after 7 days of storage, micelles were still effective enough to encapsulate most of the drug. As for the dilution stability, the divergences among samples prepared with different solvents were remarkable. The appearance of samples diluted with saline and 5% glucose solution was obviously different and the micelles were demulsified within 20 min after dilution. Yet, the micelle performed with distilled water was stable for more than 3 days. These results revealed the excellent stability of the micelle, which would satisfy the requirements of the pharmacokinetic study.

Dilution stability is an important parameter for assessing a drug delivery system since one of the major differences between in vitro and in vivo conditions is the dilution effect after administration. The drug-loaded micelles are usually significantly diluted and dissociated into monomers once the carrier reaches the circulation. Therefore, a drug delivery system with great dilution stability provides number of advantages [16]. It has been shown that the enhancement of dilution stability of mixed micelles can be attributed to the presence of phospholipids [17]. In this study, the mixed micelles were formed after mixing and stirring the phospholipid and Cremophor EL using a suitable ratio. The system provided a larger hydrophobic cargo space for DS compared with the micelles made from separate surfactants. The DS spontaneously participated in micelle cores from the aqueous environment thus increasing the micelle solubility. The mixed micelles have a higher solubilizing ability due to the existence of their hydrophobic cores, which enhance their stability upon dilution [18].

### 2.4. In Vitro Dissolution Test

In vitro dissolution tests of drug-loaded micelles and raw DS under sink conditions were performed using the dialysis method in phosphate buffer (pH 6.8, 500 mL) to assess the increase in the drug dissolution rate [19]. The results are shown in Figure 3D. Our data indicated that the cumulative percentage release of pure DS was no more than 25% even after 24 h, while the value of DS-loaded micelles was higher than 35% over 60 min; and the total amount of DS released was more than 75% after a period of 12 h. The micelles in the current study showed a more rapid release and a higher dissolution rate of DS than raw material. These results revealed that both the solubility and dissolution rate of DS could be effectively improved by the mixed micelle carrier.

### 2.5. Hemolysis Test

In order to assess the blood compatibility of the mixed micelles, an in vitro hemolysis assay method was preformed according to the design presented in Table 1. The supernatant was achromatic and transparent, and cells were re-dispersed in the tubes containing micelles. In addition, the hemolysis rate of micelle with concentration of 1 mg/mL was no more than 2.5%, indicating a slight hemolysis. According to the American Society for Testing and Materials, biomaterials can be classified into non-hemolytic (0–2% hemolysis), slightly hemolytic (2–5% hemolysis) and hemolytic (>5% hemolysis) [20]. The hemolytic test results indicated that the DS micelle at a concentration of 1 mg/mL had a good compatibility and was suitable for intravenous injection.

### 2.6. Allergenicity Test

The allergic reactions and symptoms observed in guinea pigs 20 and 120 min after injection are listed in Table 2. No allergic reactions were observed in the saline group. In the Cremophor EL injection group, half of the animals sowed an allergic reaction of Grade 3/+++ in 20 min, while only two guinea pigs and one guinea pig showed Grade 2/++ allergic responses in the blank micelle group and drug-loaded micelle group after administration. After a longer period (120 min after injection), stronger allergic reactions appeared. The guinea pigs treated with Cremophor EL showed a 100% positive reaction, and 16.7% extremely positive reactions leading to death. However, the other groups had 66.7% incidence of positive reaction and no incidence of extremely positive reaction. Even though the positive reaction of the micelle groups was as high as 66.7%, the allergic reactions in the Cremophor EL group was significantly more serious than in the micelle groups. The results showed that the potential for allergic reactions was reduced by incorporating Cremophor EL into mixed micelles.

### 2.7. Pharmacokinetics of DS-M

The plasma concentration-time profiles of DS after intravenous administration of DS-M and DS are shown in Figure 3E. The related pharmacokinetic parameters of DS in the two formulations are listed in Table 3. The plasma concentration of DS delivered by the mixed micelles was four times higher than pure DS during the experimental period. In addition, the T_max_ was sharply decreased compared with the raw DS. Moreover, the AUC_(0-t)_ of DS in micelle was 4.5 times higher than that of pure DS. These data were consistent with those of dissolution study and they indicated a better bioavailability of DS.

## 3. Materials and Methods

### 3.1. Materials

DS was purchased from Zhuhai Free Trade Zone Lizhu Synthetic Pharmaceutical Co., Ltd. (Zhuhai, China). Phospholipid was bought from Shanghai TyweiCo., Ltd. (Shanghai, China). Cremophor EL was purchased from Dalian Guanghui Technologies Corporation, Ltd. (Shenyang, China). Pyrene was bought from Sigma (St. Louis, MO, USA). Water was double distilled. Acetonitrile and methanol were of HPLC grade, and other reagents were of analytical grade.

### 3.2. Preparation of Mixed Micelles

The mixed micelles were prepared by magnetic stirring-ultrasonic dispersion. Briefly, phospholipid (1 g) was dissolved in anhydrous ethanol (3 mL). The obtained solution (0.3 mL) was then mixed with Cremophor EL (1.2 mL) and stirred for several seconds. Anhydrous ethanol (0.5 mL) was added to get a system with a total volume of 2 mL. Consequently, DS (20 mg) was added to the mixture, which was stirred at 1200 rpm for 15 min using a magnetic stirrer. Afterwards, an orange-yellow dispersion was obtained after magnetically stirring for 10 min at room temperature. The blank mixed micelles were prepared following the steps listed above.

### 3.3. Characterization of DS-M

#### 3.3.1. Size Determination and Morphological Features

Micelles were diluted with distilled water and their size and zeta potential were measured by Dynamic Light Scattering (DLS) using a Zetasizer (Malvern Nano ZSP, Malvern, UK) at room temperature. Before the measurements, the instrument was calibrated using a standard sample. The morphology of the mixed micelles was observed by Transmission Electron Microscopy (TEM, H 7650, Hitachi, Tokyo, Japan). The samples for TEM measurement were prepared according to the following procedure: a drop of mixed micelle sample after dilution was pipetted onto a carbon–coated copper for several minutes. The excess liquid was removed by filter paper and a thin liquid film was formed. The film on the grid was negatively stained with a droplet of 2% (*w*/*v*) phosphotungstic acids for 1 min. After the excess solution was removed, the sample was air-dried at room temperature for 5 min before measurement.

#### 3.3.2. Measurement of Encapsulation Efficiency (EE)

The EE was measured according to previously described approach [21]. Briefly, DS-M diluted 10 fold with distilled water was filtered through a 0.22 μm nylon (polyamide) membrane and 0.1 mL of the filtered liquid was accurately transferred into a volumetric flask containing 2 mL of methanol. After the structure of micelle was destroyed (ultrasound for 20 min), the sample was diluted to 5 mL with methanol and the DS containing in the micelle was determined using a HPLC method. The quantification of DS was performed using an Agilent 1100 series HPLC system (Agilent, Santa Clara, CA, USA), which was equipped with the Agilent 1200 series DAD detector and a reversed-phase C18 column (4.6 mm × 250 mm, 5 μm, Diamonsil plus). The data were captured and processed using Agilent ChemStation for LC 3D systems acquisition software. The mobile phase was a mixture of methanol, acetonitrile and water (pH = 3.0, 3:3:4, *v*/*v*) and eluted at a flow rate of 1.0 mL/min. Effluents were detected at 385 nm. This method has been validated for selectivity, linearity, limit of detection and quantification, accuracy, precision, as well as repeatability. The EE was determined using the following equation: EE (%) = weight of drug encapsulated in micelles/ weight of drug added × 100.(1)

#### 3.3.3. Determination of the CMC

The CMC value of the mixed micelles was measured using a pyrene fluorescence method [21]. A stock solution of pyrene (6.08 mg/L) was prepared in benzene and stored at 4 °C until use. In detail, blank micelles were prepared as described above and diluted with distilled water. Pyrene stock solution (0.1 mL) was loaded into 25 mL vials and the benzene evaporated overnight. Micelle solutions with a series of different concentrations was respectively added to the corresponding vials and the samples were incubated at room temperature for 24 h. Fluorescence excitation spectra were obtained by a microplate reader (Synergy H1, Biotek, VT, USA). Intensity ratio of excitation spectra at I1 (intensity of first peak) to I3 (intensity of third peak) was determined and plotted as a function of polymer concentration. The CMC value for each micelle solution was identified at the inflection point of the plot [22].

### 3.4. Stability of the Micelle

Stability of storage and dilution were evaluated in this study. For the storage stability, samples of drug-loaded micelles were prepared and stored at room temperature. At the time of 1, 2, 3, 5, 7 days, the samples were diluted 10 folds with distilled water and filtered through 0.22 μm nylon membrane. Their size and EE were then analyzed and compared.

For dilution stability, samples were diluted with distilled water, saline and 5% glucose solution, respectively. After 2 h, the diluted samples were filtered through a 0.22 μm nylon (polyamide) membrane filter and the remaining DS in the solution was assayed by HPLC.

### 3.5. In Vitro Dissolution Test

In vitro dissolution study was performed in phosphate buffer (Ph = 6.8, 37 ± 0.5 °C) consisting of 0.2 M monopotassium phosphate and 0.2 M sodium hydrate. A total of 5 mL of diluted micelle (containing 5 mg of DS) was transferred into a dialysis bag (MWCO5000) with two end seals. The experiment was performed on a RCZ-8B dissolution tester (Tianda Tianfa Technology Co., Ltd., Tianjin, China) with a paddle rotation at 100 revolutions per minute (rpm). At predetermined interval (0.1, 0.2, 0.3, 0.5, 1, 2, 4, 6, 8, 10, 12, 24 h), 5 mL of the release medium was removed from sample and replaced with 5 mL of fresh dissolution medium. The DS content in the solution was assayed by HPLC to determine its dissolution rate.

### 3.6. Hemolysis Test

To investigate the hemolytic potential of the mixed micelle and ensure its safety, hemolysis tests on erythrocytes were preformed [23]. Briefly, fresh rabbit blood (10 mL) was obtained. After the fibrinogen was removed by stirring with a glass rod, the obtained rabbit blood cells were rinsed several times until supernatants were colorless. The obtained cells were diluted with aqueous phosphate-buffered saline (PBS) solution (1:50, *v*/*v*) to obtain a 2% erythrocyte standard dispersion that was stored at 4 °C.

Different volumes (0.1, 0.2, 0.3, 0.4 and 0.5 mL) of mixed micelles diluted to a DS concentration of 1 mg/mL were added into tubes containing 2.5 mL of 2% erythrocyte dispersion. Adequate amounts of 1× PBS solution were added in every tube to obtain a final volume of 5 mL. The 0% and 100% hemolysis controls were obtained with 2.5 mL of 1× PBS solution and 2.5 mL of distilled water, respectively. The obtained mixtures were incubated at 37 °C for 2 h and then placed in an ice bath for 5 min to stop the hemolysis reaction. The supernatants after centrifugation at 2000 rpm for 10 min were obtained and its absorbance was determined with a microplate reader (Synergy H1) at 575 nm. The hemolysis rate was calculated according to the following equation:HR (%) = (Abs − Abs_0_)/(Abs_100_ − Abs_0_)(2)
where Abs is the absorbance of DS-loaded micelle; Abs_100_ is the hemolysis sample treated with distilled water; Abs_0_ is the PBS group absorbance value, which is the negative control.

### 3.7. Allergenicity Test

For allergenicity testing, guinea pigs were selected. Twenty four (24) guinea pigs, weighing between 280–300 g were divided into four groups (*n* = 6): a control group, a blank micelle treatment group, a DS-loaded micelle treatment group and a Cremophor EL treatment group. The samples were diluted with distilled water to obtain a concentration of 1.2 mg/mL. In the drug-loaded micelle treatment group, the formulations were administered at a dosage of 5.5 mg/kg. The control group and blank micelle group were given the same volume of corresponding solution separately. The Cremophor EL group was administrated the same volume as the volume of Cremophor EL contained in the solution given to the DS-loaded micelle treatment group. After administration, the performance of animals was monitored and ranked according to Table 2.

### 3.8. Pharmacokinetics of Micelles in Rats

#### 3.8.1. Animals

Twelve male Sprague-Dawley rats, weighing from 180–200 g were purchased from Beijing Vital River Laboratory Animal Technology Co., Ltd. (Beijing, China). All the animals were housed in an environment with a temperature of 22 ± 1 °C, relative humidity of 50 ± 1% and a light/dark cycle of 12/12 h. In addition, all rats had free access to food and water. Before administration, all rats were fasted for 12 h but had free access of water. All animal studies (including the euthanasia procedures) were done in compliance with the regulations and guidelines of Chinese Academy of Medical Sciences University institutional animal care and conducted according to the AAALAC and the IACUC guidelines.

#### 3.8.2. Chromatographic Conditions

A LC/MS/MS method was utilized to determine the DS concentration in plasms. The system was composed of a Triple Quad Mass Spectrometer (Agilent Corp.) and a HPLC column (Durashell C18, 2.1 mm × 50 mm, 1.8 μm). Mass spectrometry was performed using a negative ionization mode (ESI‑) electrospray ionization interface and quantitative analysis was performed using multiple reaction monitoring (MRM). The ion pairs used for monitoring were 315.1→200.04 (DS) and 229.1.00→185.06 (internal standard, resveratrol), respectively. The mobile phase consisted of solvent A (0.1% aqueous formic acid Solution) and solvent B (0.1% aqueous formic acid in methanol). Analytes were eluted with the following gradient program: gradient elution started at 5% B in 1.5 min and linearly increased to 95% from 1.5 min to 8 min, and then decreased to 5% from 11.5 min to 15 min. The flow rate was 0.25 mL/min and the column temperature was 35 °C. The injection volume was 20 μL.

#### 3.8.3. Pharmacokinetic Study and Sample Preparation

Twelve SD rats were randomly divided into two groups. One group (*n* = 6) received a tail vein injection of 10 mg/kg DS-loaded micelle, while the other group (*n* = 6) was administered with DS raw material at the same dose. After administration, blood samples were collected via the plexus venous in the eye ground into the heparinized tubes at different pre-determined intervals (0, 5, 15, 30 min and 1, 2, 4, 6, 8, 10, 12 and 24 h post-injection). All the blood was immediately centrifuged at 5000 rpm for 10 min. The supernatant plasma samples were obtained and stored at −80 °C until further analysis. A liquid–liquid phase extraction procedure with methanol was used for the extraction of DS from plasma. After thawing the frozen plasma samples, 300 μL of methanol was added into the mixture of 15 μL of resveratrol solution and 100 mL of plasma sample, and the resulting mixture was vortexed for 3 min at room temperature. After centrifugation at 12,000 rpm for 10 min, the organic layer was collected, and then 200 μL were transferred to a clear tapered centrifuging tube and evaporated under nitrogen at 37 °C. The residue was re-dissolved in 100 μL methanol and 20 μL of the supernatant were injected into the system.

## 4. Conclusions

In this study, we have developed a mixed micelle system based on phospholipid-Cremophor EL for injection delivery of DS. The mixed micelles composed of phospholipid and Cremophor EL were stable enough for storage and dilution. The amount of Cremophor EL used in this prescription was significantly reduced and the allergic reactions caused by the micelles was less than that caused by pure Cremophor EL. The pharmaceutics data indicated that the micelles were useful in improving the bioavailability of DS. To sum up, the described mixed micelles are a promising drug delivery system, which can be used to improve the solubility and bioavailability of DS.

## Figures and Tables

**Figure 1 molecules-24-00728-f001:**
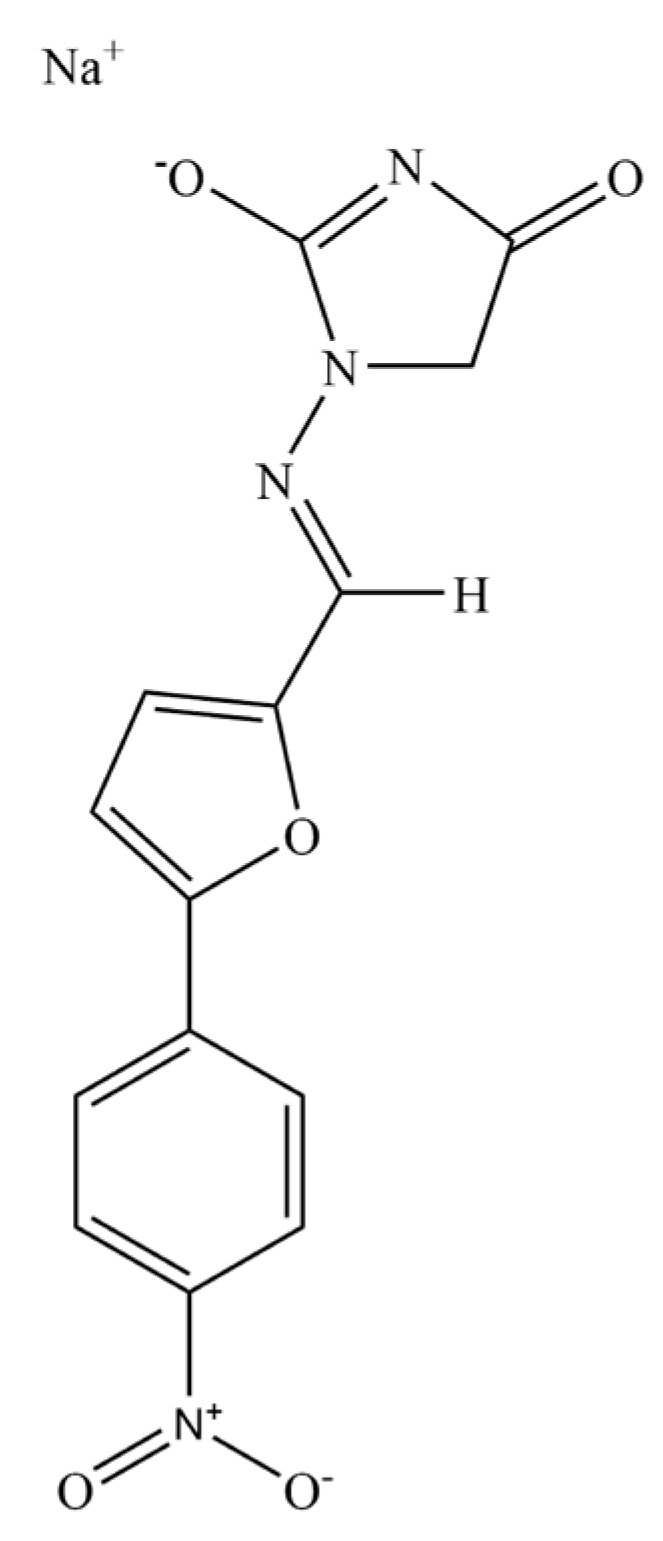
Structure of dantrolene sodium.

**Figure 2 molecules-24-00728-f002:**
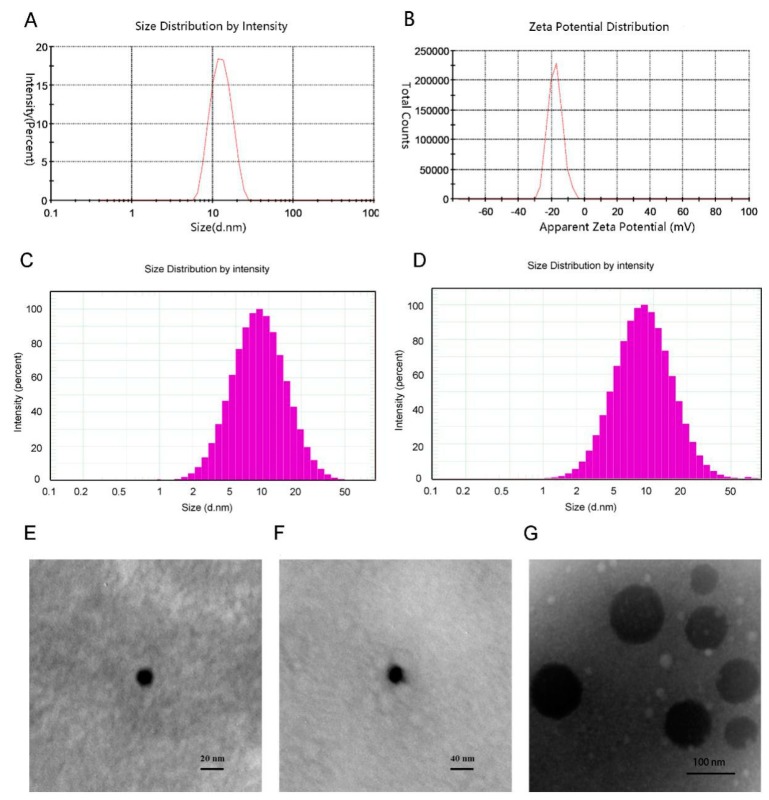
(**A**) Micelle size and size distribution of DS-M. (**B**) Zeta potential of DS-M. (**C**,**D**) Histograms representing micellar size distribution of drug-free micelle (**C**) and drug-loaded micelle (**D**). (**E–G**) Transmission electron microscopic (TEM) image of blank micelle (**E**), of DS-M (**F**), and of phospholipid (**G**).

**Figure 3 molecules-24-00728-f003:**
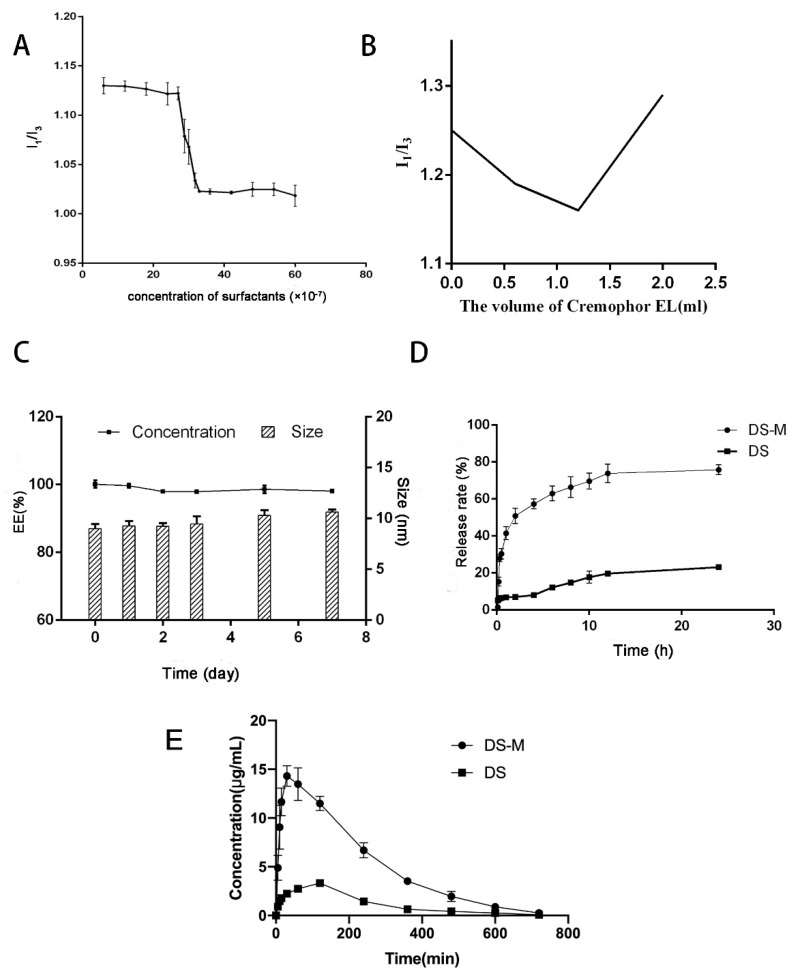
(**A**) The I1/I3 ratio of DS-M prepared by the ideal formulation. (**B**) The I1/I3 ratios of micelle with different volume of Cremophor EL. (**C**) Stability study of the micelle: the changes in DS encapsulation efficiency (%) and micelle size (nm). (**D**) In vitro dissolution test: the release rate-time curves of DS-M and pure DS. (**E**) Pharmacokinetics study: mean plasma concentration–time curves of DS-M and DS in rats after i.v. administration of a dose of 10 mg/kg DS (mean ± SD, *n* = 6).

**Table 1 molecules-24-00728-t001:** Hemolysis rate.

	Number						
	1	2	3	4	5	6	7
Micelle (mL)	0.1	0.2	0.3	0.4	0.5	-	-
PBS (mL)	2.4	2.3	2.2	2.1	2.0	2.5	-
Distilled water (mL)	-	-	-	-	-	-	2.5
2% Erythrocyte dispersion (mL)	2.5	2.5	2.5	2.5	2.5	2.5	2.5
Hemolysis rate (%)	1.25	1.37	2.17	2.30	2.28	0	100

In vitro hemolysis rate (HR) of the DS-M (*n* = 3).

**Table 2 molecules-24-00728-t002:** Levels of allergic reaction symptoms of the four groups of animals at 20 and 120 min.

Time	Group	N	Level of Allergic Reaction Symptoms in 20 min					Incidence of Positive Reaction	Incidence of Extremely Positive Reaction
			-	+	++	+++	++++		
20 min	I	6	6	0	0	0	0	0	0
	II	6	0	2	1	3	0	100%	0
	III	6	4	2	0	0	0	33.3%	0
	IV	6	5	1	0	0	0	16.7%	0
120 min	I	6	6	0	0	0	0	0	0
	II	6	0	0	2	3	1	100%	16.7%
	III	6	2	2	2	0	0	66.7%	0
	IV	6	2	3	1	0	0	66.7%	0

Levels of allergic reaction symptoms of the four groups at 20 and 120 min. “-” = negative intensity, for normal symptom of normal. “+” = intensity of weakly positive, for the symptoms of restlessness, piloerection, tremor, nosal scratch. “++” = intensity of positive, for the symptoms of sneezing, cough, tachypnea, urination, diaphoresis, lacrimation. “+++” = intensity of strongly positive, for the symptoms of dyspnea, rales, purpura, gait instability, leap, gasping, spasm, horizontal turn, tidal respiration. “++++” = intensity of extremely strong positive, for the symptoms of death. I for control group. II for Cremophor EL treatment group. III for blank micelle treatment group. IV for dantrolene sodium-loaded micelle treatment group.

**Table 3 molecules-24-00728-t003:** Pharmacokinetic parameters of DS in DS-loaded micelle and pure DS.

Parameters	DS-M	DS
C_max_(μg/mL)	13.26 ± 1.29	3.14 ± 0.16
T_1/2_(min)	169 ± 34	130 ± 21
T_max_(min)	33 ± 17	116 ± 23
AUC_(0-t)_ (μg/min/mL)	3529 ± 57	792 ± 21
AUC_(0-∞)_ (μg/min/mL)	3716 ± 39	842 ± 35

Pharmacokinetic parameters of DS in DS-M and pure DS (*n* = 6). Cmax: maximum plasma concentration. Tmax: time to reach maximum plasma concentration. T_1/2_: half time. AUC: the area under the concentration-time curve.

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
