# Peer review of "A Novel Dantrolene Sodium-Loaded Mixed Micelle Containing a Small Amount of Cremophor EL: Characterization, Stability, Safety and Pharmacokinetics"

_molecules, 2019, doi:10.3390/molecules24040728_

Round 1
Reviewer 1 Report
In this manuscript, the authors. A novel dantrolene sodium-loaded mixed micelle
containing a small amount of Cremophor EL: characterization, stability, safety and
pharmacokinetics: is a very interesting work with 4 sections. 1. Introduction. 2. Materials ad methods. 3. Results and discussion. 4. Conclusions.
However, I cannot recommend this manuscript to be published in this version in molecules due to the issues as follows.
Line 133: The authors described the encapsulation efficiency but the total of the moles or mg encapsulated in the miceles is misssing. Also the quantity of the Cremophor EL is not indicated
Line 146. The critical micelle concentration (CMC) is not indicated in the text
Line 157: "The remained" should be "the remained"
Line 164: For clarity please describe what is "ZCZ-8B"
Line 167: For clarity please describe what is PBS s
Line 190: "Abs is the absorbance of the sample" which is the sample " the sample is the DS-loaded micelle, the Cremophor EL or the DS. Please clarify.
Line 202. In allergic test the authors used the control, the Cremophor EL, the micelle treatment group and dantrolene sodium-loaded micelle. But, where are the data for only the dantrolene sodium. I think that is necessary to give the data to compare the efficiency of the micelle.
Line 270: 0.3 ml 0f phospholipid ethanol should be " 0.3 ml of phospholipid ethanol"
Line 285: The combination use of phospholipid and Cremophor EL decreased the amount required for establish an ideal system of micelle. Please can you indicate the amount of the moles or gr of each of the components to obtain the best system because it not showed in the manuscript.
Line 297: The size of the micelles is in function of the concentration? or is in dependence of the number of moles of the DS. Please clarify
Figure 2. a) Intensity (Percent) and b) Total counts are wrong direction
Please include histograms representing micellar size distribution of drug-free and drug loaded s to see the differences between the components
References
Please check the references all need to in the same format.
Author Response
Dear professor, We appreciate your valuable comments very much, which are very helpful to improver the quality of our present study. According to the comments, we have revised our paper as followings: Point 1. Line 133: The authors described the encapsulation efficiency but the total of the moles or mg encapsulated in the micelles is missing. Also the quantity of the Cremophor EL is not indicated Response 1: The encapsulation efficiency is an important index to evaluation the drug-loaded capability of carrier which is defined as the follow formulation: EE = the amount of drug encapsulated in the micelle/the total amount of drug that added. The encapsulated amount of drug is determined with HPLC and the EE is obtained. Point 2. Line 146. The critical micelle concentration (CMC) is not indicated in the text. Response 2: This part is the description of method and the CMC value is presented in part of 3.2 , line 310. Point 3. Line 157: "The remained" should be "the remained" Response 3: Thank you for your kind reminder and I have correct the spelling mistake. Point 4. Line 164: For clarity please describe what is "ZCZ-8B" Response 4: It is so sorry for making a spelling mistake. The “RCZ-8B” is the model of the dissolution tester manufactured by Tianda Tianfa Technology Co., Ltd.. It has been described in the part of 2.5, line 164. Point 5. Line 167: For clarity please describe what is PBS. Response 5: Phosphate-buffered saline (abbreviated PBS) is a buffer solution commonly used in biological research and I have describe it according to your instruction. Point 6. Line 190: "Abs is the absorbance of the sample" which is the sample " the sample is the DS-loaded micelle, the Cremophor EL or the DS. Please clarify. Response 6: Thank you for you kind reminder and I have clarified that the sample is the DS-loaded micelle in part 2.6, the corresponding line 190 . Point 7. Line 202. In allergic test the authors used the control, the Cremophor EL, the micelle treatment group and dantrolene sodium-loaded micelle. But, where are the data for only the dantrolene sodium. I think that is necessary to give the data to compare the efficiency of the micelle. Response 7: The probability of allergic reaction is duo to the introduction of Cremophor El into the mixed micelle and the design of allergic test is in order to evaluate the safety of the carrier. The efficiency of micelle is presented by the comparison of group 3 with group 1 and that of dantrolene sodium is obtained from group 4 and group 3. Point 8. Line 270: 0.3 ml 0f phospholipid ethanol should be " 0.3 ml of phospholipid ethanol" Response 8: Thank you very much for pointing out my spelling mistake and I have corrected it. Point 9. Line 285: The combination use of phospholipid and Cremophor EL decreased the amount required for establish an ideal system of micelle. Please can you indicate the amount of the moles or gr of each of the components to obtain the best system because it not showed in the manuscript. Response 9: The amount of the moles or gr of each of the components is presented in part of 3.1, line 267. The ideal system of micelle is consist of 20 mg of DS, 0.3 ml of0f phospholipid ethanol solution(250 mg/ml), 1.2 ml of Cremophor EL and 0.5 ml of dehydrated ethanol. Point 10. Line 297: The size of the micelles is in function of the concentration? or is in dependence of the number of moles of the DS. Please clarify Response 10: The size of the micelles is in function of vary factors such as concentration, the kind of surfactants, temperature and so on. It is difficult to make it clear what influence the micelle size. Point 11. Figure 2. a) Intensity (Percent) and b) Total counts are wrong direction Response 11: Thank you for your kind reminder and the mistake has been correct. Point 12. Please include histograms representing micellar size distribution of drug-free and drug loaded s to see the differences between the components. Response 12: As reviewer suggested that the histograms representing micellar size distribution of drug free and drug-loaded micelle were presented in the Figure 3 C and D. The size of DS-M is a little larger than drug-free micelle while the difference is not remarkable. Point 13. Please check the references all need to in the same format. Response 13: We have review this part and make sure the references in the same format.
Reviewer 2 Report
Manuscript of Jin et al. with the title: A novel dantrolene sodium-loaded mixed micelle containing a small amount of Cremophor EL: characterization, stability, safety and pharmacokinetics is very important and interesting, contains many experiments. Authors study Dantrolene sodium (DS) incorporation in the mixed micele, which improve the DS pharmacokinetics profile. Mixed micelles are not expensive such as polimeric micele.
Suggestions:
In the mixed micelle studies it should bee give the molar ratio of surfactants in their binary mixture (phosfolipids and EL) from which get the mixed micelle. It can changes this ratio and to determined CMC of every binary mixture, with the goal to find which surfactants ratio give the CMC with the lowest value.
In the Figure 3 A. The x axe should be the overall surfactant concentration (concentration of binary mixed surfactants mixture with the known molar ratio of surfactants), but not the concentration only EL.
Author Response
Dear professor, We appreciate your valuable comments very much, which are very helpful to improver the quality of our present study. According to the comments, we have revised our paper as followings:
Point 1. In the mixed micelle studies it should been give the molar ratio of surfactants in their binary mixture (phospholipids and EL) from which get the mixed micelle. It can changes this ratio and to determined CMC of every binary mixture, with the goal to find which surfactants ratio give the CMC with the lowest value.
Response 1: Thank you for pointing this out. After receiving your suggestion, we added the supplementary test and results were shown in Figure and table. In the supplementary experiment, the volume ratio of Cremophor EL and phospholipids was changed which their total volume kept as 2 ml. With the change of Cremophor EL volume the I1/I3 varied and the corresponding CMC values changed too. The lowest CMC value come up to the formulation we selected above. In the Figure 3 B, the I1/I3-Cremophor EL volume curve was presented and the CMC value was prescribed in part 3.2 Characterization of DS-M, line 296 .
Point 2. In the Figure 3 A. The x axe should be the overall surfactant concentration (concentration of binary mixed surfactants mixture with the known molar ratio of surfactants), but not the concentration only EL.
Response 2: We are so sorry for making such a spelling mistake and we have corrected it according to your suggestion.

Reviewer 3 Report
The manuscript by Jin et al is focused on preparation and the characterization (TEM morphology, hydrodynamic parameters, zeta-potential) and pre-clinical study (stability, safety, and pharmacokinetics) of dantrolene sodium-loaded mixed micelle containing phospholipids and Cremophor EL. Despite numerous English language errors, the paper is presented in a clear, logical manner, and the experiments are performed in details. However, the choice of Molecules MDPI for the publication of this paper remains unclear to me (no chemistry presented in this paper). In my opinion, Pharmaceuticals MDPI or another relevant journal would be a more appropriate choice for this paper. I leave this issue to the discretion of the Editor.
Specific comments:
· Please add clear and complete legends to all Figures and Tables. Each legend should contain as much information as possible about what the Table or Figure tells the reader.
· Table 3: The standard deviation should be expressed as ONE significant figure; that is, unless the number is between 11 and 19 times some power of ten, in which case you can use two significant figures. The mean value should be rounded off at the decimal place corresponding to the last significant digit of its standard deviation. E.g., 102.45 ± 5.65 should be presented as 102 ± 6; 3727.58 ± 17.82 should be presented as 3727 ± 18 etc.
· Figure 3 A and B: The ordinate is not DS concentration; please correct.
· Remove lines 287, 442, 456-457
· There are too many English errors and wrong constructions, which sometimes drastically affect the scientific meaning, making the paper difficult of reading and understanding. A native English speaker with a scientific background should carefully revise the manuscript prior to its resubmission.
Author Response
Dear professor,
We appreciate your valuable comments very much, which are very helpful to improver the quality of our present study. According to the comments, we have revised our paper as followings:
Point 1.
Please add clear and complete legends to all Figures and Tables. Each legend should contain as much information as possible about what the Table or Figure tells the reader.
Response 1: We are sorry for our negligence of the information and we have added the following legends to all Figures and tables and the corresponding explanations were added too.
Point 2.
Table 3: The standard deviation should be expressed as ONE significant figure; that is, unless the number is between 11 and 19 times some power of ten, in which case you can use two significant figures. The mean value should be rounded off at the decimal place corresponding to the last significant digit of its standard deviation. E.g., 102.45 ± 5.65 should be presented as 102 ± 6; 3727.58 ± 17.82 should be presented as 3727 ± 18 etc.
Response 2: We are sorry for the negligence of the standard deviation and we have correct the date according to your suggestion.
Point 3.
Figure 3 A and B: The ordinate is not DS concentration; please correct.
Response 3: Thank you for your kind reminder and we have made correction according to your suggestion.
Point 4.
Remove lines 287, 442, 456-457
Response 4: We have removed lines 282, 442, 456-457.
Point 5.
There are too many English errors and wrong constructions, which sometimes drastically affect the scientific meaning, making the paper difficult of reading and understanding. A native English speaker with a scientific background should carefully revise the manuscript prior to its resubmission.
Response 5: We tried our best to improve the manuscript and made some changes in the manuscript. These changes will not influence the content and framework of the paper. And we have marked the changes in the revised paper.

Round 2
Reviewer 1 Report
The authors. A novel dantrolene sodium-loaded mixed micelle containing a small amount of Cremophor EL: characterization, stability, safety and pharmacokinetics: is a very interesting work. However, I cannot recommend this manuscript to be published in this version in molecules because is full of grammatical mistakes. My recommendation is find an English specialist
Line 92. Figure 1. structure of dantrolene sodium should be "Structure of ......."
Line 278. "water to obtained" should be "water to obtain"
Line 316. "The mobile phase consist" should be "The mobile phase consists"
Line 350. "ethanol solution(250 mg/ml)" should be " ethanol solution (250 mg/ml)"
Line 407. "Creomphor EL" should be "Creomphor EL"
Line 409. The figures has a mistakes (figure 3E)
Line 420. i.v. "administration" "Administration"
Line 471. "can" "could"
Commentary about the figure 3B are missing in the text.
Line 484. "Materials" should be "materials"
Line 506. Should be revised
Line 549. should be references
Author Response
Dear professor,
Thank you very much for your comments concerning our manuscript entitled “A novel dantrolene sodium-loaded mixed micelle containing a small amount of Cremophor EL: characterization, stability, safety and pharmacokinetics”. Those comments are all valuable and very helpful for revising and improving our paper, as well as the important guiding significance to our researches. We have studied comments carefully and have made correction which we hope meet with approval.
We tried our best to improve the manuscript and made some changes in the manuscript. An English specialist was invited to polish the paper and the grammatical mistakes have been corrected. These changes will not influence the content and framework of the paper. And here we did not list the changes but marked them in revised paper.
We appreciate for your warm work earnestly, and hope that the correction will meet with approval.

Reviewer 3 Report
The authors have successfully addressed all the reviewers’ concerns, improving the manuscript with their edits.
Author Response

(The authors gave the same response as above.)
